

# Underrepresentation of women in the senior levels of Brazilian science

Jaroslava V. Valentova[1], Emma Otta[1], Maria Luisa Silva[2] and Alan G. McElligott[3,4]

[1] Department of Experimental Psychology, Institute of Psychology, University of Sao Paulo, Sao Paulo, Brazil
[2] Instituto de Ciências Biológicas, Laboratório de Ornitologia e Bioacústica, Universidade Federal do Pará, Belém, Pará, Brazil
[3] Queen Mary University of London, Biological and Experimental Psychology, School of Biological and Chemical Sciences, London, United Kingdom
[4] Current affiliation: Department of Life Sciences, University of Roehampton, London, United Kingdom

Corresponding authors
Jaroslava V. Valentova,
jaroslava@usp.br
Alan G. McElligott,
alan.mcelligott@roehampton.ac.uk

## ABSTRACT

Despite significant progress, there is still a gender gap in science all over the world, especially at senior levels. Some progressive countries are recognizing the need to address barriers to gender equality in order to retain their best scientists and innovators, and ensure research excellence and social and economic returns on the investment made by taxpayers each year on training women scientists. We investigated the gender distribution of: (i) the productivity scholarship (PS) holders of the Brazilian National Council for Scientific and Technological Development (Conselho Nacional de Desenvolvimento Científico e Tecnológico, CNPq, $N = 13,625$), (ii) the members of the Brazilian Academy of Science (Academia Brasileira de Ciências, ABC, $N = 899$), and (iii) the amount of funding awarded for top quality research ("Universal" Call of CNPq, $N = 3,836$), between the years of 2013 and 2014. Our findings show evidence for gender imbalances in all the studied indicators of Brazilian science. We found that female scientists were more often represented among PS holders at the lower levels of the research ranking system (2). By contrast, male scientists were more often found at higher levels (1A and 1B) of PS holders, indicating the top scientific achievement, both in "Engineering, Exact Sciences, Earth Sciences", and "Life Sciences". This imbalance was not found in Humanities and Social Sciences. Only 14% of the ABC members were women. Humanities and Applied Social Sciences had a relatively low representation of women in the Academy (3.7%) compared to Engineering, Exact and Earth Sciences: 54.9% and Life Sciences: 41.4%. Finally, female scientists obtained significantly more funding at the lower level of the research ranking system (2), whereas male scientists obtained significantly more funding at the higher levels (1A and 1B). Our results show strong evidence of a gender imbalance in Brazilian science. We hope that our findings will be used to stimulate reforms that will result in greater equality in Brazilian science, and elsewhere.

## INTRODUCTION

Although some progress has been made, there is evidence of a persistent gender gap in science (*Shen, 2013*) (see a special 2013 issue of the journal *Nature:* http://www.nature.com/women; *Martin, 2014*; *McAllister, Juillerat & Hunter, 2016*). All over the world, many women are deterred from pursuing a career in science to the most senior levels. Women scientists are still promoted less, win fewer grants, and are more likely to leave research than similarly qualified men. Whereas around half of those who gain doctoral degrees in science and engineering in the United States and Europe are women, only one-fifth of full professors are women (*Handelsman et al., 2005*).

Women represent less than 12% of the membership of the world's national science academies, according to a study conducted by the Academy of Science of South Africa (ASSAf; *Academy of Science of South Africa-ASSAf, 2016*). An analysis of 69 National Science Academies revealed that the average proportion of members who were women stood at just 12% in 2013–14. Women were relatively more represented in the Academies of North America, Latin America and the Caribbean, with the Cuban Academy of Sciences having the highest proportion of female members (27%). Examining the membership of the science academies of Brazil, China, France, and the USA according to gender, it was found that the percentage of women varies between 6 and 14% (China 6%, France 11%, USA 13%, Brazil 14%; *Academy of Science of South Africa-ASSAf, 2016*).

Some progressive countries have recognized the need to address barriers to gender equality in order to retain their best scientists and innovators, and to ensure research excellence. Policies that focus on gender equality in science would also assist with making sure that social and economic returns on the investments made by taxpayers each year on training women scientists are not wasted. The Athena SWAN Charter is an accreditation and improvement program for higher education and research organizations focusing on gender and other forms of inequality, which was established in the UK in 2005 (*Equality Challenge Unit (ECU), 2016*; *Munir, 2014*). This program is proving successful in improving gender equality in terms of promotion and retention of women in science (*Munir, 2014*). Similarly, in Australia, the Science in Australia Gender Equity (SAGE) was created in 2015 in order to address problems related to gender equality (*Science in Australia Gender Equity (SAGE), 2016*). SAGE is an initiative of the Australian Academy of Science in partnership with the Australian Academy of Technological Sciences and Engineering, and was adapted from the Athena SWAN Charter (*Munir, 2014*).

Innovative national policies that affect research funding include gender balance targets (e.g., Slovenia, Switzerland) and legislation on gender quotas of up to 40% of the minority gender on committees (e.g., Finland, Norway). Some countries also have policies to increase university funding based on their performance in terms of gender equality (e.g., Ireland, Germany and Netherlands) (*Boyle et al., 2015*).

### Gender gap in Brazilian science

Recently, a large study aimed at research performance (mainly production of papers) over past 20 years in a number of world regions conducted by *Elsevier (2017)* reported that during the period between 2011 and 2015 nearly half (49%) of Brazilian scientific studies

were produced by women. This is in a sharp contrast with the period of 1996–2000, when women contributed with only 38% of the Brazilian scientific production. Thus, together with Portugal (that showed similar recent changes), Brazil is one of the countries closing the gender gap in scientific production.

However, although Brazilian women publish recently as much as men, and although some scientific areas (for example, Biology and Humanities) are represented more by women students (*Varella et al., 2016*), the gender ratio among the top scientific positions tells a different story. One way how to investigate gender composition among the top professional scientific positions in Brazil is to analyse the distribution of productivity scholarships provided by the Brazilian National Council for Scientific and Technological Development (Conselho Nacional de Desenvolvimento Científico e Tecnológico—CNPq).

CNPq is a funding agency of the Brazilian federal government. It was created in 1951 and is dedicated to the promotion of scientific and technological research, as well as the creation of human resources for research (*Chaimovich & Melcop, 2007*; *Oliva & Da Silva, 2012*). It also plays a major role in the formulation and implementation of science, technology and innovation policies in Brazil. CNPq offers several levels of highly competitive scholarships, such as monthly scholarships for students at graduate and postgraduate degrees, and also Productivity Scholarships (PS) aimed at "researchers who stand out among their peers, according to their scientific production". Among the candidates, CNPq selects proposals for financial support that are likely to contribute significantly to the scientific and technological development and innovation in any area of knowledge for Brazil. The requests are evaluated by 66 committees within the three big areas of CNPq equivalent to STEM Science with more than 300 members. Researchers are classified into Category 1 (at least eight years after receiving PhD) and Category 2 (at least three years after receiving PhD). Within Category 1, a researcher is classified in one of four different levels (1A, 1B, 1C or 1D), based on his/her scientific output (peer-reviewed journal articles, books and book chapters), scientific coordination of research networks, and contribution to human resources creation (theses supervisions of MSc and PhD students, and supervision of undergraduate research students) during the previous 10 years. The four levels present a decreasing order of prestige and value. Thus, the researcher 1A is classified at the highest possible range of scientific research according to the ranking system of the country. There is no level specification for category 2, and the researcher's productivity is evaluated by his/her publications and student supervisions during the last five years. All researchers start at the category 2 level, and ask for reclassification after three years of their work. The classification between category 1 and 2, which depends on years after receiving PhD, does not take into account maternity leave (four or six months) or any other career breaks.

There is a lot of research on Productivity Scholarship Holders (PSH) of CNPq, focused on specific sub-areas of knowledge. Recent studies in Chemistry (*Alves, Yanasseb & Somac, 2014*), Odontology (*Cavalcante et al., 2008*), Physical Education (*Leite et al., 2012*), Veterinary Medicine (*Spilki, 2013*) and Medicine (*Melo & Casemiro, 2003*; *Mendes, Martelli & Souza Filho, 2010*; *Martelli-Junior et al., 2010*) show that PSH researchers are predominantly men (60–76%). In these areas, men also receive the majority (around 56%) of Category 2 grants. In Pediatrics (*Gonçalves et al., 2014*) and Cardiology (*Oliveira*

*et al., 2011*), a similar pattern was found, as in the whole of Medicine. One decade earlier, when a slightly different system was used by CNPq to classify PS researchers (1A, 1B, 1C, 2A, 2B, 2C), the profile of researchers in public health also showed a clear gender gap (*Barata & Goldbaum, 2003*). Whereas 1A (PhD > 21 years), 1B (PhD > 20 years) and 1C (PhD > 14 years) researchers were typically men, 2A (PhD > 12 years) could be men or women, and 2B (PhD > 9 years) and 2 C (PhD > 6 years) were typically women (*Barata & Goldbaum, 2003*).

An analysis of CNPq's database in 2002 revealed that Brazilian women made up a larger proportion of junior student positions than men (e.g., undergraduate Scientific Initiation Scholarships and Masters Scholarships), but were less represented than men in the most senior professional roles (e.g., Productivity Scholarships; *Leta, 2003*): Scientific Initiation Scholarship (55% of 14,040), Masters Scholarships (53% of 5,592), PhD Scholarships (48% of 5,734), Postdoctoral Fellowships (50% of 376), Productivity Scholarships (32% of 7,763). A proportion of the women that go through the early stages of scientific careers are therefore "lost" along the way, or simply do not get the recognition from peers to continue to conduct research with grants (*Leta, 2003*). However, the gender gap seems to be reversed in some fields. For example, in 2013 among Psychology the majority of PSH researchers were women (63%) across all the ranking levels (*Wendt et al., 2013*).

Another way to study gender differences among Brazilian scientists is to analyse the gender composition of the Brazilian Academy of Science (Academia Brasileira de Ciências—ABC). The ABC is a prestigious honorific scientific society, founded in 1916 (*Oliva & Da Silva, 2012*). The most important representatives of the Brazilian scientific community are admitted as members after a rigorous selection process. They have a leadership role in the advancement of scientific and technological activities of the country. ABC interacts with the federal government and its agencies, identifying research priorities in several issues of national interest, focused on the economic development and the well-being of the population. This contributes to new programs and actions of the national policy of Science, Technology and Innovation. The supra-institutional nature of ABC, the leadership role and ample diversity in the areas of interests of its members enable ABC to discuss and propose new solutions to scientific and socio-economic issues that require a multidisciplinary approach.

## AIMS OF THE CURRENT STUDY

Here, we studied the gender distribution of: (a) the productivity scholarship holders (PSH) of the Brazilian National Council for Scientific and Technological Development (CNPq; *Conselho Nacional de Desenvolvimento Científico e Tecnológico (CNPq), 2016a*), (b) the members of the Brazilian Academy of Science (ABC; *Academia Brasileira de Ciências (ABC), 2016*), and (c) the amount of funding awarded for research ("Universal" Call of CNPq; *Conselho Nacional de Desenvolvimento Científico e Tecnológico (CNPq), 2016b*). We analysed the data according to gender, area of knowledge, and level in the research ranking system. More specifically, the goal of the present research was to offer a more comprehensive picture of gender imbalances in Brazilian science, by comparing the areas of Engineering, Exact Sciences, Earth Sciences, Life Sciences, and the Humanities and

Applied Social Sciences. According to previous research, we expected a higher frequency of male researchers among scholarship holders and among the Academy of Science members. We also hypothesized that the gender gap would be less pronounced in the fields of Humanities and Social Sciences compared to Exact Sciences. The overarching goal of our research is to raise awareness of gender imbalance in science and stimulate to future policy changes that address will this problem.

## MATERIALS AND METHODS

The study sample included a total 13,625 productivity scholarship holders (PSH) of the Brazilian National Council for Scientific and Technological Development (CNPq); 4,859 from the area of Engineering, Exact Sciences, and Earth Sciences (ETEC), 5,687 from the area of Life Sciences (LS), and 3,079 from the area of Humanities and Applied Social Sciences (HASS), according to lists publicly available at the site of the agency at January 2016 (*Conselho Nacional de Desenvolvimento Cientifico e Tecnológico (CNPq), 2016a*). The study sample also included a total of 899 active members of the Brazilian Academy of Sciences according to lists organized by gender publicly available at the site of ABC at January 2016 (*Academia Brasileira de Ciĕncias (ABC), 2016*). Their area of knowledge was identified according to Lattes *curricula* (unified system of academic information for Brazilian researchers and students), photos were checked, and CNPq research categories (1A, 1B, 1C, 1D, and 2) were determined for those who were productivity scholarship holders.

We analysed an additional sample of 3,836 researchers awarded funding for research in the UNIVERSAL MCTI/CNPQ CALL (No 14/2014), comparing the results in the three funding ranges by gender: less than 30 thousand Brazil reals (BRL; 3.12 BRL = 1 US dollar, USD), 30–60 thousand BRL and 60–120 thousand BRL. The call was open to all scientific disciplines and types of research, from basic research to applied research, including scientists who are not PS holders. Researchers with PhD > 7 years ago could apply for the lower funding range (<BRL 30,000 = USD 9,619). The intermediate funding range (<BRL 60,000 = USD 19,240) was open to Level 2 researchers. Level 1 PS holders could only submit proposals to the higher funding range (<BRL 120,000 = USD 38,480).

### Statistical analysis

Separate datasets were created for Engineering, Exact Sciences and Earth Sciences (ETEC), Life Sciences (LS), and Humanities and Applied Social Sciences (HASS) productivity scholarship holders, for researchers awarded funding and for members of the Brazilian Academy of Science. The datasets were constructed with information on gender, sub-area of knowledge, level of research category (1A, 1B, 1C, 1D, 2) and amount of funding. Categorical (dichotomous or nominal) variables were compared using the chi-square test in SPSS 20.0.

To explore the possible effect of generation on gender differences, we analysed age distribution of the ABC members (full members, associated members and others ranged from 30 to 102 years), with the mean of 66.05 years (SD = 15.96). The histogram showed two peaks in the age distribution, at approximately 40 and 70 years old. Thus, we divided

the sample into two categories, 30–50 ($N = 178$), and more than 50 ($N = 779$), to explore whether there might be a generation effect on the gender distribution.

We used a Crosstabs procedure to test the independence of two categorical variables. If the chi-square was significant, indicating an association among the variables, but the table was larger than $2 \times 2$, we requested the adjusted standardized residuals from among the options in the Cells dialog (or/CELLS subcommand). Adjusted residuals are interpreted in the following way: under the null hypothesis that the two variables are independent, the adjusted residuals will have a standard normal distribution, i.e., have a mean of 0 and standard deviation of 1. Therefore, an adjusted residual that is more than 1.96 (2.0 is used by convention) indicates that the number of cases in that cell is significantly larger than would be expected if the null hypothesis was true, with a significance level of .05. An adjusted residual that is less than $-2.0$ indicates that the number of cases in that cell is significantly smaller than would be expected if the null hypothesis were true. Thus, the standardized residual shows whether there are fewer or more cases (depending on the sign of the adjusted residual) than would be expected if the two variables were independent. It is worth noting that the results are not based on expectation of 50% of women in each category, but calculation of each case of the expected value is based on the total of rows, total of columns and on the general $N$ according to the chi-square test.

## RESULTS

For the CNPq's productivity scholarship holders, we found that women are involved in all areas of research, but equally represented only in Humanities and Applied Social Sciences (50%, $\chi 2 = 0.094$, $N = 3,079$, $df = 1$, $p = 0.75$), compared with Life Sciences (41%, $F\chi 2 = 171.992$, $N = 5,687$, $df = 1$, $p < 0.001$) and Engineering, Exact and Earth Sciences (20%, $\chi 2 = 1739.17$, $N = 4,859$, $df = 1$, $p < 0.001$).

### Gender distribution of productivity scholarship holders in sub-areas of knowledge

There are significantly fewer women than expected by chance in some sub-areas of Engineering, Exact and Earth Sciences, ($\chi 2 = 267.05$, $df = 21$, $p < 0.001$). The adjusted residuals revealed that the number of women was smaller than expected in the sub-areas of Biomedical Engineering, Electrical Engineering, Mathematics, Mechanical Engineering, and Physics, but was greater than expected in Chemical Engineering, Chemistry, Industrial Design, Materials Engineering, Nuclear Engineering, Oceanography Production Engineering and Sanitary Engineering (Table 1, for details see Table 2).

A significant association was found between gender and the sub-areas of Life Sciences (CV), ($\chi^2 = 788.70$, $df = 29$, $p < 0.001$). The number of women was smaller than expected by chance in Agricultural Engineering, Agronomy, Biophysics, Fisheries Engineering, Forest Engineering, Physical Education, Veterinary Medicine, Zoology, Zootechnology, but was greater than expected in Biochemistry Botany, Food Science and Technology, Genetics, Immunology, Microbiology, Morphology, Nursing, Nutrition, Pharmacology, Pharmacy, Phonoaudiology, Physiotherapy and Public Health (Table 1, see also Table 3).

**Table 1** Distribution of the productivity scholarship holders by gender and sub-areas of Engineering, Exact and Earth Sciences (W, Women; M, Men; n, frequencies; AR, Adjusted residuals).

| Sub-areas | W (n) | M (n) | Total n | AR W |
|---|---|---|---|---|
| Chemistry | 207 | 480 | 687 | **7.1** |
| Chemical engineering | 59 | 95 | 154 | **5.7** |
| Industrial design | 14 | 16 | 30 | **3.6** |
| Nuclear engineering | 25 | 48 | 73 | **3.0** |
| Materials engineering | 87 | 247 | 334 | **2.8** |
| Sanitary engineering | 36 | 88 | 124 | **2.5** |
| Production engineering | 26 | 62 | 88 | **2.2** |
| Oceanography | 33 | 85 | 118 | **2.2** |
| Computer science | 88 | 287 | 375 | 1.7 |
| Geosciences | 107 | 363 | 470 | 1.5 |
| Probability and statistics | 19 | 51 | 70 | 1.5 |
| Transportation engineering | 13 | 39 | 52 | 0.9 |
| Civil engineering | 56 | 210 | 266 | .4 |
| Astronomy | 20 | 79 | 99 | .0 |
| Aerospace engineering | 10 | 44 | 54 | −.3 |
| Mining engineering | 4 | 21 | 25 | −.5 |
| Marine engineering | 1 | 10 | 11 | −.9 |
| Biomedical engineering | 4 | 60 | 64 | **−2.8** |
| Mathematics | 29 | 271 | 300 | **−4.7** |
| Mechanical engineering | 24 | 252 | 276 | **−4.9** |
| Electrical engineering | 13 | 269 | 282 | **−6.7** |
| Physics | 101 | 806 | 907 | **−7.5** |
| Total | 976 | 3,883 | 4,859 | |

There was also a significant association between gender and the sub-areas of Humanities and Applied Social Sciences (HASS), ($\chi 2 = 360.06$, $df = 23$, $p < 0.001$). The number of women was smaller than expected by chance in Administration, Economics and Philosophy, but was greater than expected in Arts, Education, Information Science, Linguistics, Psychology and Social Services (Table 1, see also Table 4).

## Distribution of productivity scholarship holders by gender and scholarship level

There was a significant association between gender and scholarship level in Engineering, Exact Sciences and Earth Sciences ($\chi 2 = 45.70$, $df = 4$, $p < 0.001$), Life Sciences (CV; $\chi 2 = 89.20$, $df = 4$, $p < 0.001$), and Humanities and Applied Social Sciences (CHSA; $\chi 2 = 13.78$, $df = 4$, $p < 0.01$). In ETEC and LS, the number of women was greater than expected in the lower level of the research ranking system, whereas the number of men was greater than expected in the higher levels (1A and 1B; Table 4). A different pattern was found in HASS, with a smaller gender gap in the research ranking system (Table 4).

In Engineering, Exact Sciences and Earth Sciences, the proportion of men was higher in Chemistry ($\chi 2 = 16.38$, $df = 4$, $p < 0.01$), Civil Engineering (Fisher's Exact Test = 16.24, $p < 0.01$), and Sanitary Engineering (Fisher's Exact Test = 9.89, $p < 0.05$). In Life

**Table 2 Distribution of the productivity scholarship holders by gender and sub-areas of Life Sciences (W, Women; M, Men; n, frequencies; AR, Adjusted residuals).**

| Subareas | W (n) | M (n) | Total n | AR W |
|---|---|---|---|---|
| Nursing | 165 | 8 | 173 | 14.7 |
| Phonoaudiology | 50 | 1 | 51 | 8.3 |
| Nutrition | 54 | 27 | 81 | 4.7 |
| Public health | 114 | 85 | 199 | 4.7 |
| Microbiology | 105 | 82 | 187 | 4.2 |
| Genetics, | 134 | 115 | 249 | 4.1 |
| Botanics | 115 | 95 | 210 | 4.0 |
| Immunology, | 90 | 69 | 159 | 4.0 |
| Physiotherapy | 43 | 23 | 66 | 4.0 |
| Pharmacy | 88 | 68 | 156 | 3.9 |
| Food science and technology | 99 | 82 | 181 | 3.7 |
| Pharmacology | 102 | 87 | 189 | 3.6 |
| Morphology | 64 | 52 | 116 | 3.1 |
| Biochemistry | 113 | 119 | 232 | 2.3 |
| Physiology | 86 | 92 | 178 | 1.9 |
| General biology | 4 | 1 | 5 | 1.8 |
| Parasitology | 66 | 77 | 143 | 1.2 |
| Odontology | 82 | 129 | 211 | −.7 |
| Medicine | 205 | 333 | 538 | −1.6 |
| Aquaculture | 22 | 47 | 69 | −1.6 |
| Ecology | 68 | 126 | 194 | −1.8 |
| Biophysics | 24 | 59 | 83 | −2.3 |
| Fisheries engineering | 28 | 79 | 107 | −3.2 |
| Zoology | 64 | 157 | 221 | −3.8 |
| Veterinary medicine | 91 | 208 | 299 | −3.9 |
| Physical education | 14 | 70 | 84 | −4.6 |
| Forest engineering | 26 | 121 | 147 | −5.9 |
| Zootechnics | 59 | 195 | 254 | −6.0 |
| Agricultural engineering | 17 | 127 | 144 | −7.3 |
| Agronomy | 157 | 604 | 761 | −12.4 |
| Total | 2,349 | 3,338 | 5,687 | |

Sciences the proportion of men was higher in Medicine ($\chi 2 = 23.78$, $df = 4$, $p < 0.001$), Public Health ($\chi 2 = 18.85$, $df = 4$, $p < 0.001$), Physiology (Fisher's Exact Test $= 26.22$, $p < 0.001$), Agronomy (Fisher's Exact Test $= 20.67$, $p < 0.05$), Pharmacology (Fisher's Exact Test $= 18.40$, $p < 0.001$), Odontology (Fisher's Exact Test $= 11.82$, $p < 0.05$), and Botany (Fisher's Exact Test $= 10.50$, $p < 0.05$). In Humanities and Applied Social Sciences, significant associations between gender and scholarship level were found for Psychology (Fisher's Exact Test $= 61.76$, $p < 0.001$) and Urban Planning (Fisher's Exact Test $= 15.26$, $p < 0.01$), with more women than expected by chance in the intermediate levels of the research ranking system, PS-1B and PS-1C.

**Table 3** Distribution of the productivity scholarship holders by gender and sub-areas of the Humanities and Social Sciences (W, Women; M, Men; Total $n$, frequencies; AR, Adjusted residuals).

| Sub-areas | W ($n$) | M ($n$) | Total $n$ | AR W |
|---|---|---|---|---|
| Linguistics | 152 | 59 | 211 | **6.7** |
| Social service | 62 | 9 | 71 | **6.4** |
| Education | 242 | 136 | 378 | **5.9** |
| Information science | 35 | 10 | 45 | **3.8** |
| Psychology | 175 | 138 | 313 | **2.3** |
| Art | 61 | 42 | 103 | **2.0** |
| Letters | 126 | 102 | 228 | 1.7 |
| Urban planning | 44 | 33 | 77 | 1.3 |
| Architecture and urbanism | 54 | 42 | 96 | 1.3 |
| Domestic economy | 1 | 0 | 1 | 1.0 |
| Anthropology | 74 | 66 | 140 | .8 |
| Archaeology | 23 | 19 | 42 | .7 |
| Tourism | 8 | 6 | 14 | .6 |
| Communication | 61 | 61 | 122 | .1 |
| History | 113 | 125 | 238 | −.7 |
| Geography | 40 | 51 | 91 | −1.1 |
| Theology | 2 | 5 | 7 | −1.1 |
| Sociology | 88 | 106 | 194 | −1.3 |
| Museology | 1 | 4 | 5 | −1.3 |
| Law | 26 | 42 | 68 | −1.9 |
| Political science | 42 | 77 | 119 | −3.2 |
| Management | 50 | 126 | 176 | −5.8 |
| Philosophy | 22 | 111 | 133 | −7.8 |
| Economics | 29 | 178 | 207 | −10.6 |
| Total | 1,531 | 1,548 | 3,079 | |

## Gender distribution of the Brazilian academy of science

There were 126 female members (14%) of the Brazilian Academy of Science (ABC; $\chi2 = 465.64$, $N = 899$, $df = 1$, $p < 0.001$). Examining the gender distribution as a function of areas of knowledge, using the classification of the National Council for Scientific and Technological Development (CNPq), the gender gap is more pronounced in Engineering, Exact and Earth Sciences (8.9%), and less pronounced in Life Sciences (20.4%), and Humanities and Applied Social Sciences (18.2%), ($\chi2 = 23.87$, $N = 899$, $df = 1$, $p < 0.001$). Male scientists predominate in all areas of knowledge, and the representation of Humanities and Applied Social Sciences in the academy is low (Table 5).

The percentage of women in the Brazilian Academy of Sciences does not correspond to the percentage of women among Productivity Scholarship Holders; it is two times lower. This occurs in all areas of knowledge (Fig. 1). For Brazilian Academy of Science members who were Productivity Scholarship Holders, there was a significant association between gender and scholarship level in Life Sciences (Fisher's Exact Test = 10,08, $p < 0.05$), but not in Engineering, Exact Sciences and Earth Sciences ($\chi2 = 4,765$, $df = 4$, NS). The

**Table 4** Distribution of the productivity scholarship (PS) holders by gender and scholarship level (W, Women; M, Men; n, frequencies; AR, Adjusted residuals.)

| Level | W (n) | M (n) | Total n | W (%) | M (%) | AR W |
|---|---|---|---|---|---|---|
| **Exact sciences and earth sciences** | | | | | | |
| PS-1A | 41 | 378 | 419 | 9.8% | 90.2% | −5.5 |
| PS-1B | 75 | 391 | 466 | 16.1% | 83.9% | −2.3 |
| PS-1C | 78 | 380 | 458 | 17.0% | 83.0% | −1.7 |
| PS-1D | 161 | 595 | 756 | 21.3% | 78.7% | 0.9 |
| PS-2 | 621 | 2,139 | 2,760 | 22.5% | 77.5% | 4.8 |
| Total | 976 | 3,883 | 4,859 | 20.1% | 79.9% | |
| **Life sciences** | | | | | | |
| PS-1A | 132 | 393 | 525 | 25.1% | 74.9% | −7.9 |
| PS-1B | 172 | 343 | 515 | 33.4% | 66.6% | −3.8 |
| PS-1C | 248 | 351 | 599 | 41.4% | 58.6% | 0.1 |
| PS-1D | 418 | 580 | 998 | 41.9% | 58.1% | 0.4 |
| PS-2 | 1,379 | 1,671 | 3,050 | 45.2% | 54.8% | 6.4 |
| Total | 2,349 | 3,338 | 5,687 | 41.3% | 58.7% | |
| **Humanities and applied social sciences** | | | | | | |
| PS-1A | 126 | 143 | 269 | 46.8% | 53.2% | −1 |
| PS-1B | 158 | 132 | 290 | 54.5% | 45.5% | 1.7 |
| PS-1C | 154 | 113 | 267 | 57.7% | 42.3% | 2.7 |
| PS-1D | 219 | 262 | 481 | 45.5% | 54.5% | −2 |
| PS-2 | 874 | 898 | 1,772 | 49.3% | 50.7% | −0.5 |
| Total | 1,531 | 1,548 | 3,079 | 49.7% | 50.3% | |

**Table 5** Distribution of the members of the Brazilian Academy of Science by gender and area of knowledge (W, Women; M, Men; n, frequencies; AR, Adjusted residuals).

| Area | W (n) | M (n) | Total n | W (%) | M (%) | AR W |
|---|---|---|---|---|---|---|
| Engineering, exact sciences and earth | 44 | 450 | 494 (54.9%) | 8.9 | 91.1 | −4.9 |
| Life sciences | 76 | 296 | 372 (41.4%) | 20.4 | 79.6 | 4.7 |
| Humanities and applied social sciences | 6 | 27 | 33 (3.7%) | 18.2 | 81.8 | .7 |
| Total | 126 | 773 | 899 | | | |

small sample size in Humanities and Applied Social Sciences precludes detailed analyses. However, in LS, the number of women was greater than expected in the lower level of the research ranking system (1D), whereas the number of men was greater than expected in the higher level (1A, Table 6).

When divided into two age categories (30–50 and >50 years), there was a significant gender difference in both age categories in Engineering, Exact and Earth Sciences, and Life Sciences, and a significant difference between men and women in Humanities and Applied Social Sciences (Table 7). When we compared the proportions of women between the younger and the older category, we found a significant difference in the total number of women ($\chi 2 = 22.35$, $df = 1$, $p < 0.001$), and also specifically in Engineering, Exact and Earth Sciences ($\chi 2 = 8.22$, df $= 1$, $p < 0.001$), Life Sciences ($\chi 2 = 11.87$, $df = 1$, $p < 0.001$),

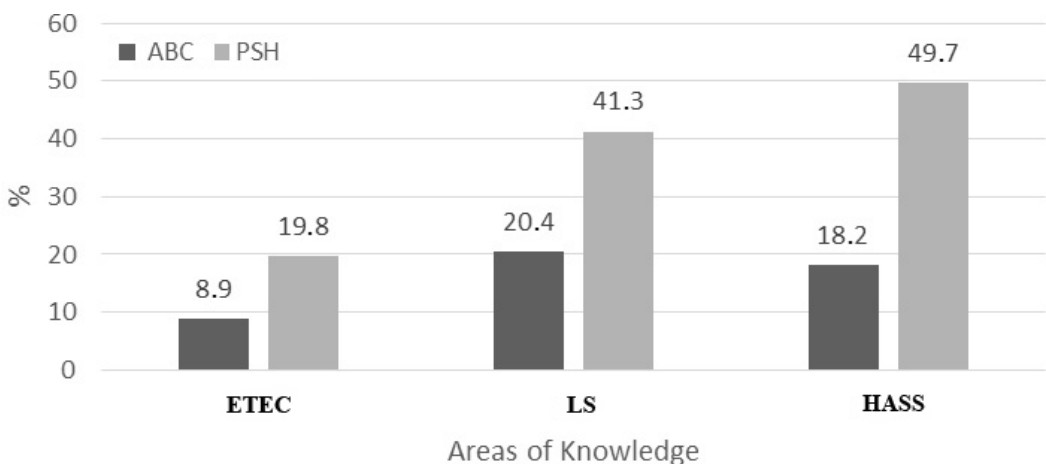

**Figure 1** **Comparative distribution of female researchers who are members of the Brazilian Academy of Science (ABC—dark bars) and Productivity Scholarship Holders (PSH—light bars) by areas of knowledge (ETEC, Engineering, Exact Sciences and Earth Sciences; LS, Life Sciences; HASS, Humanities and Applied Social Sciences).** The reverse percentages are applicable for male researchers.

**Table 6** **Distribution of the members of the Brazilian Academy of Science who are Productivity Scholarship Holders by gender and scholarship level (W, Women; M, Men; *n*, frequencies; AR, Adjusted residuals).**

| Level | W (*n*) | M (*n*) | Total *n* | W (%) | M (%) | AR W |
|---|---|---|---|---|---|---|
| **Exact sciences and earth sciences** | | | | | | |
| PS-1A | 17 | 137 | 154 | 11.0% | 89.0% | 0.7 |
| PS-1B | 3 | 27 | 30 | 10.0% | 90.0% | .0 |
| PS-1C | 0 | 13 | 13 | 0.0% | 100% | −1.2 |
| PS-1D | 4 | 18 | 22 | 18.2% | 81.8% | 1.4 |
| PS-2 | 2 | 42 | 44 | 4.5% | 95.5% | −1.3 |
| Total | 26 | 237 | 263 | | | |
| **Life sciences** | | | | | | |
| PS-1A | 20 | 99 | 119 | 16.8% | 83.2% | −2.8 |
| PS-1B | 4 | 13 | 17 | 23.5% | 76.5% | .0 |
| PS-1C | 7 | 11 | 18 | 38.9% | 61.1% | 1.6 |
| PS-1D | 9 | 13 | 22 | 40.9% | 59.1% | **2.0** |
| PS-2 | 8 | 17 | 25 | 32.0% | 68.0% | 1.0 |
| Total | 48 | 153 | 201 | | | |
| **Humanities and applied social sciences** | | | | | | |
| PS-1A | 4 | 4 | 8 | 50.0% | 50.0% | .8 |
| PS-1B | 0 | 0 | 0 | – | – | |
| PS-1C | 0 | 1 | 1 | | 100% | −.9 |
| PS-1D | 0 | 1 | 1 | | 100% | −.9 |
| PS-2 | 1 | 1 | 2 | 50.0% | 50.0% | .3 |
| Total | 5 | 7 | 12 | | | |

**Table 7  Percentages of men and women among the members of the Brazilian Academy of Sciences (ABC) divided into a younger group (30–50 years) and an older group (more than 50 years).**

| Age categories | Total $N$ (%) | | Engineering, exact and earth sciences | | Life sciences | | Humanities and applied social sciences | |
|---|---|---|---|---|---|---|---|---|
| | Men | Women | Men | Women | Men | Women | Men | Women |
| 30–50 | 140 (78.7) | 38 (21.3) | 78 (86.7) | 12 (13.3) | 60 (70.6) | 25 (29.4) | 2 (66.7) | 1 (33.3) |
| $\chi 2$ | 58.45*** | | 48.40*** | | 14.41*** | | .33 | |
| >50 | 708 (91.1) | 69 (8.9) | 411 (94.9) | 22 (5.1) | 262 (86.5) | 41 (13.5) | 24 (82.8) | 5(17.2) |
| | 527.45*** | | 350.46*** | | 162.12*** | | 12.45*** | |

Notes.
*** $p < .001$.

**Table 8  Distribution of gender by amount of funding in the UNIVERSAL MCTI/CNPQ CALL-No 14/2014 in the major areas of knowledge: Humanities and Applied Social Sciences (HASS), Life Sciences (LS) and Engineering, Exact Sciences and Earth Sciences (ETEC). W, Women; M, Men; $n$, frequencies; AR, Adjusted residuals.**

| Amount | W ($n$) | M ($n$) | Total $n$ | W (%) | M (%) | AR W |
|---|---|---|---|---|---|---|
| **Exact sciences and earth sciences** | | | | | | |
| 60–120 thousand reais | 38 | 152 | 190 | 20.0% | 80.0% | −1,6 |
| 30–60 thousand reais | 76 | 231 | 307 | 24.8% | 75.2% | ,1 |
| Less than 30 thousand reais | 172 | 492 | 664 | 25.9% | 74.1% | 1,2 |
| Total | 286 | 875 | 1,161 | | | |
| **Life sciences** | | | | | | |
| 60–120 thousand reais | 112 | 218 | 330 | 33.9% | 66.1% | −3,3 |
| 30–60 thousand reais | 236 | 362 | 598 | 39.5% | 60.5% | −1,5 |
| Less than 30 thousand reais | 503 | 593 | 1,096 | 45.9% | 54.1% | **3,8** |
| Total | 851 | 1,173 | 2,024 | | | |
| **Humanities and applied social sciences** | | | | | | |
| 60–120 thousand reais | 25 | 37 | 62 | 40.3% | 59.7% | −1,8 |
| 30–60 thousand reais | 86 | 78 | 164 | 52.4% | 47.6% | ,4 |
| Less than 30 thousand reais | 222 | 203 | 425 | 52.2% | 47.8% | ,8 |
| Total | 333 | 318 | 651 | | | |

while there was no difference in Humanities and Applied Social Sciences ($\chi 2 = 0.45$, $df = 1$, $p = 0.50$).

## Amount of funding awarded by gender

Analysing the amount of funding awarded by gender in the Universal CNPq Call, it was found that female scientists obtained more funding at the BRL < 30,000 range, whereas male scientists obtained more funding at the higher ranges, especially BRL < 120,000 (contingency chi-square test, $\chi 2 = 24.20$, $N = 3,836$, $df = 2$, $p < 0.001$). An additional analysis of the amount of funding awarded by gender in the Universal CNPq Call, conducted separately by areas of knowledge (Table 8), showed that the association was significant in Life Sciences ($\chi 2 = 17.195$, $N = 2,024$, $df = 2$, $p < 0.001$), but not in Humanities and Applied Social Sciences, ($\chi 2 = 3.218$, $N = 651$, $df = 2$, NS), nor in Exact Sciences and Earth Sciences, ($\chi 2 = 2.277$, $N = 1,161$, $df = 2$, NS).

## DISCUSSION

Our findings show gender imbalances in senior levels of Brazilian science, considering scientific productivity scholarships, grants and membership of Brazilian Academy of Sciences. Our analysis of a sample of 13,625 productivity scholarship holders (PSH) of the Brazilian National Council for Scientific and Technological Development (CNPq) showed that female scientists were more represented among PSH at the lower level of the research ranking system, whereas male scientists were more represented at higher senior levels (1A or 1B). This was evident in Engineering, Exact and Earth Sciences, and in Life Sciences. This result is in line with the gender gap in science found at senior levels worldwide (*Shen, 2013*; *Handelsman et al., 2005*). In Brazil, the gender gap in science was reduced in Humanities and Social Science compared to Engineering, Exact and Earth Sciences, which has also been found in other countries such as the UK (*Boyle et al., 2015*).

Our analysis of a sample of 3,836 researchers awarded funding by the Universal CNPq Call shows that female scientists were more frequently awarded lower amounts to conduct their research than male scientists. We did not have access to the amounts of funding requested by gender, and therefore we do not know if women applied for similar amounts. However, it is notable that there were no significant differences in the amount of funding awarded according to gender both in Humanities and Applied Social Sciences, and in Engineering, Exact and Earth Sciences, and Life Sciences. The smaller amount of funding awarded to female scientists was only found in Life Sciences (<30,000 BRL versus <120,000 BRL). In Biomedical sciences, women also get smaller grants than men in the US (*Pohlhaus et al., 2011*) and the UK (*Bedi, Van Dam & Munafo, 2012*).

There are relatively more women members of the Brazilian Academy of Science (ABC) compared to other academies of science (*Academia Brasileira de Ciěncias (ABC), 2016*). Nevertheless our study revealed several asymmetries in ABC. For example, the proportion of men is greater in all areas of knowledge (Engineering, Exact and Earth Sciences: 8.9% women (W) versus 91.1% men (M), Life Sciences: 20.4% W versus 79.6% M; Humanities and Applied Social Sciences: 18.2% W versus 81.8% M). The proportion of researchers working in Humanities and Applied Social Sciences is very low in the ABC (3.7%) in comparison to Engineering, Exact and Earth Sciences, and Life Sciences (54.9% and 41.4%, respectively). The proportion of women in the ABC is approximately half that of women among Productivity Scholarship Holders in all areas of knowledge (Exact Sciences and Earth Sciences: 8.9% versus 19.8%; Life Sciences: 20.4% versus 41.3%; Humanities and Applied Social Sciences: 18.2 versus 49.7%). Finally, among members of the ABC who were Productivity Scholarship Holders of the area of Life Sciences, the number of women was greater than expected in the lower level of the research ranking system (1D), whereas the number of men was greater than expected in the higher level (1A). As can be expected, most ABC members are holders of higher levels of PS, and thus the differences between men and women points to the general finding of gender imbalance in the senior academic positions. Our additional analyses including age of the ABC members support this notion showing that the gender gap is smaller among a younger category of individuals.
Our results are in agreement with previous research showing that in several scientific areas, in particular Exact Sciences, women are heavily underrepresented which can be result of several factors (e.g., *Ceci & Williams, 2011*), such as women's self-exclusion from competition, and time out for maternity leave. For example, the Brazilian system of classification of scientists by CNPq does not take into account maternity leave (four or six months) or any other career breaks, which could be one of the reasons for the expected reduction in women in the higher classification levels, in particular at the beginning of their careers. In addition, the very fact of being in a minority may make women more likely to drop out. Moreover, we can speculate that scientific areas can be avoided by women because of great workloads and stress caused by such professions that is incompatible with family life. However, this would be true also for other areas, such as social sciences, veterinary medicine, medicine, or law where in some countries women dominate (e.g., *Adamo, 2013*). One of the plausible factors seems to be competitiveness and job insecurity in science (compared to medicine, for example, *Adamo, 2013*) that can be biased in many ways, such as recommendation letters (*Dutt et al., 2016*). In addition, and as supported by our data, it is important to remember that there may be generational effects; if the more senior grades are people who started careers a long time ago then there may have been different pressures and opportunities for them.

Another major reason for gender imbalances in science is that women more commonly tend to opt for areas such psychology and veterinary, while more men choose more traditional areas, such as engineering (for a meta-analysis, see *Su, Rounds & Armstrong, 2009*). This is due to many factors, such as conditioning from early ages, or cognitive differences between men and women (that can be caused by both biological and cultural factors; *Su, Rounds & Armstrong, 2009*). However, cognitive differences between men and women are quite small or even non-existent (for review see *Halpern, 2012*; *Lippa, 2005*), which means that most men and women overlap greatly in their abilities. The effect size of sex difference in career preferences is, by contrast, large (*Su, Rounds & Armstrong, 2009*), which points to the fact that differences in cognition only marginally contribute to the gender imbalance in academic representatives of many scientific fields. Thus, public opinion and popular literature exaggerating "natural" sex differences in cognition, and arbitrarily creating two distinct categories of men versus women can largely affect expectations of representation of women in science. For example, women scientists represent a minority in school science textbooks, and they are also invited less for conferences as plenary speakers (*Martin, 2014*). Changing this might start to change the public opinion that women are less competent for scientific jobs. Overall, a lack of role models and the public message of lack of competency decreases self-esteem and motivation of women to compete with men in numerous dimensions of scientific work, such as publications quantity (*Cameron, Gray & White, 2013*). As suggested by *Ceci & Williams (2011)*, it is highly important to offer to women "realistic information about career opportunities and expose them to role models" to "ensure they do not opt out of inorganic fields because of misinformation and stereotypes". Thus, early education should focus more on not biased information about scientific fields, leaving misinformation based on social stereotypes behind.

We show that there are great differences for women between Life Sciences and Engineering and Exact Sciences (ETEC), versus the Humanities and Social Science. It seems that during the career there is a significant drop of women in ETEC and most Life Sciences, while the gender gap is not much pronounced on the senior level of Humanities and Social Sciences. Again, this can be caused by many factors, similar to those discussed above. One study (*Glass et al., 2013*) compared adherence of women in STEM in general and other professional positions, showing that in many dimensions, such as family factors, these professional areas would be comparable. However, factors such as job satisfaction and in general motivation to remain active in the field, seem to be lower in STEM (*Glass et al., 2013*). Thus, the policy might focus on motivating women, and creating more women-friendly environments in STEM professions to reduce the exit rate.

Our data from the Brazilian Academy of Science contains the age distribution of their members. We were thus able to test whether the gender difference remains stable or changes through the age cohorts. We showed significantly higher proportion of women among the younger ABC members than among the older members. Thus, it seems that with lower age groups the gender difference even in the top scientific levels tends to diminish. This is in agreement with a new study showing that only recently Brazilian women publish half of the scientific papers, in comparison to the past (*Elsevier, 2017*). Although the situation seems to be improving, still the gender difference in all the scientific areas was significant among the ABC members.

The Brazilian government has programs, such as Programa Pró-Equidade de Gênero e Raça (*Secretaria de políticas para as mulheres, 2015*) (The Pro-Equity Gender and Race Program), and Observatório de Igualdade de Gênero (Brazil Observatory of Gender Equality) (*Ministério das Mulheres, da Igualdade Racial e dos Direitos Humanos, 2015*). They focus on people management and organizational culture to achieve equality between women and men in the labour market, political participation, on reduction of violence against women, but not specifically on gender inequality in science. The Brazilian Academy of Science has a leadership role in the country, contributing to the policy of Science, Technology and Innovation. The achievements of Brazilian women in science are discussed by the ABC, recognizing the importance of retaining them and making the best use of the investment made in training women scientists. Nevertheless, our research clearly shows that even with more female presence than in other countries, the imbalance is still evident.

Our research has several limitations. For example, we do not possess data on gender balance in application for Productivity Scholarships and other research grants. Thus, the present data do not show if lower numbers of women with grants are because fewer women apply, or because those who do apply are less likely to be funded. Because this would be a key factor for developing effective solutions to gender imbalance in science, future studies should focus on the difference between applicants and funded researchers.

Future research might focus not only on mapping gender differences in scientific or any other areas of human behavior, but also on specific mechanisms that lead to such gender gaps, which might help to better understand and in particularly to reduce the gender gaps. For example, *Nosek et al. (2009)* analysed Implicit Association Tests completed by citizens of 34 countries and found that nation-level implicit stereotypes associating science with

males more than with females predicted nation-level sex differences in 8th-grade science and mathematics achievement. They suggested that implicit stereotypes and sex differences in science participation and performance are mutually reinforcing, and contributing to the persistent gender gap in science engagement. Thus, the differences in top scientific levels are just the tip of an iceberg that starts as early as during childhood with different expectations that the society has from men and women.

Overt sexism has decreased over the last few decades, but subtle gender biases maybe still be exhibited by both men and women, and are held even by individuals who consider themselves as egalitarian (*Nosek, Banaji & Greenwald, 2002*; *Kite, Deaux & Haines, 2008*; *Moss-Racusina et al., 2012*). Our results show clear evidence of imbalance of gender in Brazilian science, and this probably has deep institutional and cultural roots. We agree with *Mühlenbruch & Jochimsen (2013)* that research policies are needed and only wholesale reforms will bring equality to Brazil and other countries. The Athena Swan program in the UK and similar initiatives in other countries provide important examples of how long-term progress can be achieved (*Munir, 2014*; *Science in Australia Gender Equity (SAGE), 2016*).

## ACKNOWLEDGEMENTS

This paper resulted from a talk by Dr. Alan McElligott on the UK Athena Swan gender equality program, at the XXXII Annual Meeting of Ethology in 2015, organized in Brazil by Prof. Maria Luisa da Silva.

### Funding
Financial support was provided by Capes and CNPq (BRAZIL). The funders had no role in study design, data collection and analysis, decision to publish, or preparation of the manuscript.

### Grant Disclosures
The following grant information was disclosed by the authors:
Capes and CNPq (BRAZIL).

### Competing Interests
The authors declare there are no competing interests.

### Author Contributions
- Jaroslava V. Valentova analyzed the data, contributed reagents/materials/analysis tools, wrote the paper, prepared figures and/or tables, reviewed drafts of the paper.
- Emma Otta conceived and designed the experiments, performed the experiments, analyzed the data, contributed reagents/materials/analysis tools, wrote the paper, prepared figures and/or tables, reviewed drafts of the paper.
- Maria Luisa Silva conceived and designed the experiments, performed the experiments, contributed reagents/materials/analysis tools, wrote the paper, prepared figures and/or tables, reviewed drafts of the paper.

- Alan McElligott conceived and designed the experiments, contributed reagents/materials/analysis tools, wrote the paper, reviewed drafts of the paper.

### Data Availability

The raw data has been supplied as Supplemental Dataset.

### Supplemental Information

Supplemental information for this article can be found online at http://dx.doi.org/10.7717/peerj.4000#supplemental-information.

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
