# Peer review of "Underrepresentation of women in the senior levels of Brazilian science"

_PeerJ, doi:10.7717/peerj.4000_

## Round 0.1 · original submission · Major Revisions

The reviewers have commented on aspects of the paper where greater clarification is required and I think you will find their comments helpful in indicating where some revision is needed. I have some minor comments about stylistic issues below.

My main response to this paper, however, is that the dataset has potential which is not fully exploited. You could analyse the data to give more of an indication of possible reasons for the gender differences that you describe. But I also have a concern that in places the presentation of results may be misleading.

As you note, a sharp decline in the percentage of women in science as one advances in seniority has been reported in other countries, but there is a range of explanations. These are not mutually exclusive, but include a) effects of implicit or explicit bias against women in competitive situations, b) women’s self-exclusion from competition, and time out for maternity etc. c) It has also been suggested that the very fact of being in a minority makes women more likely to drop out. d) In addition, it is important to remember that there may be generational effects: if the more senior grades are people who started careers a long time ago, then there may have been different pressures and opportunities for them. I think you should spell out these options: even though an observational study like this cannot unambiguously choose between them, there will be clues in the data.

My concern is that, at several points you make statements like “Female scientists were more frequently awarded PQ scholarships at the lower level of the system” etc (Abstract and also para 1, p 10)– this makes it sound as if the scientists apply for a PQ and then the BNC decides the level of award. My impression from your account, though, is that the specific level of award is very dependent on stage of career. It is a subtle difference, but my concern is that it is misleadingly described, as if it is the funder who decides the award level of an applicant. I suspect that, if this is similar to other countries, the applicant decides to apply for the award. This would mean that the success rate at different grades will be a function of two things: the proportion how many men and women apply at each grade, and the success rate at each grade. It really is not possible to tell from such data whether there is bias against women unless you have data on the number of applicants of each gender, and so can see if success rates differ by gender. I think this point needs to be made explicitly to avoid confusion. A related concern was the account of lower funding for women in para 2 of p 10. See point 2 below.

Despite this ambiguity, I think there are interesting features in the data which could be used to throw light on possible explanations.

1. First you have already compared patterns of gender imbalance across three broad disciplines, and these do give very different pictures. As reviewer 2 points out, the presentation of data in Table 1 is unhelpful, as what we need to see are the percentages by rows (i.e. the % female) and not by columns. This would make it easier to see that in ‘exact sciences’ you start with a low percentage of women at the lowest grades, which just declines as they become more senior. In life sciences, it starts with a fairly equal gender distribution at the lowest grades, but women again drop off as positions become more senior. This is not the case, however, for the humanities, where the initial roughly equal split between male and female at the most junior levels remains pretty constant as the grades move up in seniority. This is an important result, as it would seem to rule out any general explanation of the gender gap at senior grades being due simply to women selecting themselves out because of competing pressures of motherhood etc – these presumably would be similar in all disciplines.

2. The second opportunity that you have with these data is to consider whether the differences in grant income between genders are explicable in terms of seniority or whether there are additional biases. The most direct way to do this would be to link the information about grant level to the information about level of scholarship. I looked at your data and found you could match the names from the PQ Holders to the Edital Universal dataset for about 16% of cases – which is still a large sample size given the overall numbers. This means that, for those where PQ stage is known, you can compare income levels by gender and PQ stage, to see whether the lower income for women is explicable by their lower PQ stage. As far as I could tell from a brief look at the data, this was a sufficient explanation: i.e. it looked as if women had lower grant income because they were more likely to be at an earlier career stage. This is not entirely clearcut, because it’s unclear whether those who do not appear in the Edital database are people who applied for funding unsuccessfully or if they did not apply. But it suggests that there is no additional gender bias when funding decisions are made.

I think if you consider these further points, it would make a stronger paper. In particular, the striking differences in retention between Humanities and Life Sciences, both of which start from a similar baseline % women, raises questions about what is different for women in these two broad disciplines.

Yet another point considers the analysis of the ABC (p 8-9): if this is an elite institution, then one would predict that the people who are elected would come from the highest level of PQ, i.e. the 1A band. You note that the % of women in the ABC is lower than the % women with PQ. But if you are looking for evidence of bias, it would seem appropriate to focus on % women in band 1A . Here, it looked as if the % women in 1A and in the ABC were fairly comparable in Hard Sciences and LIfe Sciences, but the reduction in % women that you described was very pronounced in Humanities. I think this point emerges from the analysis at top of p 9, but it was a rather convoluted approach to analysis.

Overall then, I think the paper needs reshaping so you have :

1. Introduction: covers evidence for inequality and then delineates possible reasons for this
I did not find the hypotheses listed in lines 132-141 very convincing: they seemed more like redescriptions of the data. As this is essentially a descriptive study, I think it may be preferable to remove these hypotheses, but note instead that differences between subject areas may be informative about the underlying causes of gender inequality and how to tackle them.
2. Methods
3. Results (to incorporate additional analyses as above)
4. Discussion: How results inform our understanding of different causes. (see reviewer 1). You could move to Discussion the material that you currently have on p 3 paras 2-3, which focuses on possible ways of addressing gender inequality.

More minor points
Remember that you need to use full stop, rather than comma, for all decimal points. This applies to figures as well as tables.
I agree with reviewer 2 that you should put Methods before Results.
line 51: Although -> Despite
line 74: Herein -> Here
line 170: less -> fewer
line 312. Brazilian -> The Brazilian
line 316 taken literally, it sounds as if you are saying we are seeking equality between sexes in violence against women – clearly that is not so and this needs rewording
line 322-323: material is repeated and this needs rewording
p 23; I found Figure 2 very hard to understand. What do the bars represent here?
Figure 3 is probably not needed: especially as I think these data only make sense if related to career stage and discipline, both of which will affect funding.
I shared reviewer 2’s confusion about ‘adjusted residuals’. Are these the difference between expected and observed frequencies in the chi square? They are always equivalent but different in sign for male and female and it does not seem necessary to report both.

·

Basic reporting

This is an interesting submission that highlights the gender imbalance in representation of women in science in Brazil.
Minor suggestions include avoiding the use of "hard" as a descriptor of disciplines or sub-disciplines of science. Hard versus soft science? Hard versus easy science? There are some typos or corrections not incorporated (line 322-323).
Discussion could be expanded and strengthened to relate loss or exclusion of women from "higher" levels of science (but not humanities) to cultural context. Institutional, federal, local policies that support women, parents, minorities etc. Parental leave policies for instance, which differ significantly between European countries and the US leading to different stressors on women at specific stages in their scientific careers - what is the situation in Brazil? Why is the impact felt more acutely in science?. Are there recommendations that would come out of this study (along the lines of an Athena Swan or SAGE Pilot type of program for instance). The paragraph starting at 312 acknowledges some attempts at systemic change but there appears to be little effect or impact. Can the work presented here inform and guide policy-making? If so, how?

Experimental design

No comments

Validity of the findings

Data are provided and made available as per journal instructions

I would encourage the authors to be more speculative about the underlying mechanisms that lead to their observation (making it clear they are speculating as per the journal instructions). And looking to their data and findings for any potential policy recommendations (what would they be? who would be responsible?)

Reviewer 2 ·

Basic reporting

THIS IS MY FULL REVIEW OF THE PAPER
* * *
This paper presents a simple summary of gender imbalance in Brazilian science. It is not an experimental study, but rather summarises available data. If this is a reasonable topic for PeerJ, then I provide some comments to improve the paper:

Abstract - please remove or spell out acronyms (PQ, CNPq). What is meant by 'Exact Sciences' - is that mathematics? It is not a term I have come across.

Intro

I imagine the majority of readers, like me, have no familiarity with the Brazilian funding system. It would be helpful to have clearer description of the categories and the grants that can be applied for. When I first read line 87, I thought these categories were the scores applicants were given for their grant, not their career stage. It was also only later that I worked out how the level 1A/1B/1C/1D categories relates to career stage, and the level of funding. And it is not stated how this relates to doctoral or postdoctoral scholarships. It would help to set this out explicitly (maybe in a table) at an early point in the introduction.

It seems that the 1A/1B etc categories depend on years post PhD - does this calculation take into account career breaks / maternity leave etc? can the authors comment on how this might influence the gender distribution?

p131 - what does the 58% mean? Is this 58% female or 58% of the sample are category 2? the latter does not seem very relevant here.

p5 - why is this analysis divided into these particular subfields? are these categories used by the CNPq?

p6 - as far as I can tell, PeerJ does not require a 'methods at the end' format, so I strongly advise putting the methods section betwen the intro and results - it makes the paper much more coherent and easier to read.

p6 = what are the standardised residuals? how are these calculated?

p7 - the authors say there are more women than expected in some fields and less in others. but this seems to be based on the overall proportion of 25% women. Presumably, if the default expectation were 50% women, then all the subfields would be less than expected? It would be worth spelling out what the expected values mean here, so readers can make sense of the deviation from expectd.

p9 - is the analysis of level of funding confounded with career stage? It wasn't quite clear if only researchers at some career stages could apply for some funding levels, but if that is the case, then this analysis might be redundant.

Discussion

Reviewing the studies of Nosek & Moss-Racusina et al feels a bit out of place here - these are very different studies and don't really fit with the flow of this paper. It would be better to consider these studies in relation to things that could be done to reduce the gender imbalance.

It might be worth adding a limitations section - in particular, the present data doesn't show if lower numbers of women with grants are because fewer women apply, or because those who do apply are less likely to be funded. Without data on the gender balance in applications, it is impossible to determine this, but it is a key factor for developing effective solutions to gender imbalance.

It could also be worth discussing what kind of research could be done in future to better understand & reduce gender imbalances in science.

Tables 1-4

Why do the % columns sum to 100% when you read downwards? surely it would be much more helpful if they summed to 100% along the rows so it is easy to see the % women in each category?

Fig 2B seems redundant as it is just the opposite of Fig2A.


Minor grammatical issues

line 51 - "Although some progress HAS BEEN MADE, ..."

line 58-61 - rewrite this sentence

line 322-323 - rephrase

Experimental design

this is not an experimental paper so I leave it to the editor to determine if it is in scope. the analysis could be described better.

Validity of the findings

The findings seem reasonable.

Additional comments

no further comments

---

## Round 0.2 · Minor Revisions

Thank you for your careful attention to the feedback on your original submission. The PeerJ system generates the main letter, but I am pleased to tell you that I am happy to accept the paper subject to the minor revisions proposed by reviewer 2, plus attention to the following minor points:

line 225: refer to 'rows' rather than 'lines'
line 355 'in particular'
line 372 delete comma after 'areas'
lines 388-389: the quote has a type , need for -> of, ie: 'ensure they do not opt out of inorganic fields...'
line 400: environments

·

Basic reporting

The language has been improved where necessary and additional supporting information included as recommended by the reviewers. The article is professional and self-contained as per requirements of the journal.

Experimental design

Rigorous as far as I can assess.

Validity of the findings

No comment.

Additional comments

The manuscript has benefited from very useful feedback from the reviewers and is strengthened. It is a useful addition to the overall field.

Reviewer 2 ·

Basic reporting

no comment

Experimental design

no comment

Validity of the findings

no comment

Additional comments

The paper is much improved and I have just a couple of minor comments.

First, I still find the terminology confusing in places. In particular, why do you suggest that STEM is the same as 'Exact Sciences'. STEM is a term that applies to ALL science & technology (including life sciences etc). It would be more helpful to introduce the 3 domains distinguished in Brazil (Physical & exact sciences: ETEC; Life Sciences VC and Humanities and Applied Social Sciences (HASS) and then stick to those three acronyms throughout. there is no need to try to re-define STEM to mean just some sciences.

Second, I recommend caution in the sections on cognitive differences between men and women. The phrase 'cognitive differences between men and women are quite small' implies that there are still differences, when in fact many studies suggest zero difference. It is important to make this clear.

---

## Round 0.3 · accepted · Accept

Thanks for your careful attention to the minor corrections.